# xMTF: A Formula-Free Model for Reinforcement-Learning-Based Multi-Task Fusion in Recommender Systems

## Abstract

Recommender systems need to optimize various types of user feedback, e.g., clicks, likes, and shares. A typical recommender system handling multiple types of feedback has two components: a multi-task learning (MTL) module, predicting feedback such as click-through rate and like rate; and a multi-task fusion (MTF) module, integrating these predictions into a single score for item ranking. MTF is essential for ensuring user satisfaction, as it directly influences recommendation outcomes. Recently, reinforcement learning (RL) has been applied to MTF tasks to improve long-term user satisfaction. However, existing RL-based MTF methods are formula-based methods, which only adjust limited coefficients within pre-defined formulas. The pre-defined formulas restrict the RL search space and become a bottleneck for MTF. To overcome this, we propose a formula-free MTF framework. We demonstrate that any suitable fusion function can be expressed as a composition of single-variable monotonic functions, as per the Sprecher Representation Theorem. Leveraging this, we introduce a novel learnable monotonic fusion cell (MFC) to replace pre-defined formulas. We call this new MFC-based model eXtreme MTF (xMTF). Furthermore, we employ a two-stage hybrid (TSH) learning strategy to train xMTF effectively. By expanding the MTF search space, xMTF outperforms existing methods in extensive offline and online experiments. xMTF has been deployed online, serving over 100 million users.

## CCS Concepts

• **Information systems → Recommender systems**.

## Keywords

Multi-Task Fusion, Reinforcement Learning, Recommender System

**ACM Reference Format:**
Anonymous Author(s). 2025. xMTF: A Formula-Free Model for Reinforcement-Learning-Based Multi-Task Fusion in Recommender Systems. In *Proceedings of Proceedings of the ACM Web Conference 2025 (WWW '25)*. ACM, New York, NY, USA, 11 pages. https://doi.org/XXXXXXX.XXXXXXX

## 1 Introduction

Recommender systems are playing an increasingly important role in various platforms, e.g. E-commerce [8, 14, 35], videos [24, 25], news[15, 33], etc. Practical recommender systems often need to

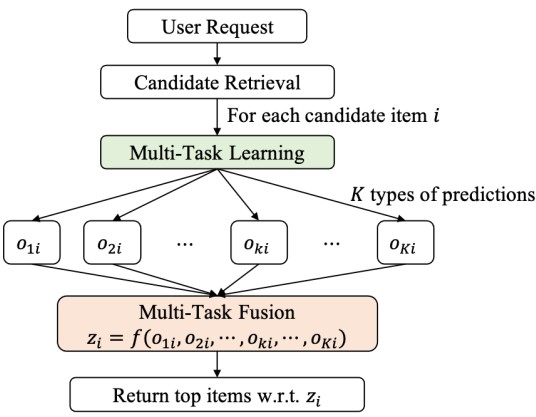

**Figure 1: MTL and MTF in recommender systems.**

optimize various types of user feedback. For instance, on video platforms, user satisfaction is influenced by watch time and interactions such as likes, follows, and shares[1, 31]. In such scenarios, recommender systems must consider multiple feedback types to deliver the final recommendation. A typical recommender system handling multiple types of feedback has two components: a multi-task learning (MTL) module, predicting feedback such as click-through rate and like rate; and a multi-task fusion (MTF) module, integrating these predictions into a single score for item ranking[32]. Figure 1 illustrates the relationship between MTL and MTF.

MTF is essential for ensuring user satisfaction with the recommender systems, as it directly affects recommendation outcomes. However, compared to MTL, which has been extensively studied [17, 23], MTF poses more challenges. MTF aims to integrate the predictions of multiple user feedback to generate a single score, reflecting the user's overall satisfaction. However, users typically do not provide direct feedback indicating overall satisfaction with each item. Instead, overall satisfaction is indicated by long-term feedback such as session length, daily watch time, and retention, which cannot be directly linked to individual recommended items. Recently, there has been growing interest in using reinforcement learning (RL) for MTF[1, 2, 4, 16, 31, 32]. RL-based approaches regard users as the environment and the recommender system as the agent, treating fusion weights as RL actions. These approaches have proven effective in MTF by modeling long-term user satisfaction.

However, existing RL-based approaches are all *formula-based MTF* approaches, i.e., to define a fusion formula $f(o_1, \cdots, o_K; a_1, \cdots a_K)$ with $K$ predictions $o_1, \cdots, o_K$ and the coefficients $\boldsymbol{a} = [a_1, \cdots, a_K]$, regarding the coefficients $\boldsymbol{a}$ as the actions of RL. Table 1 shows typical fusion formulas in recent research. By pre-defining the fusion formula, we only need to optimize a few coefficients via RL, which makes RL learning easier. Nevertheless, there are several issues with pre-defined fusion formulas. First, different formulas result in varying recommendation outcomes, making it challenging

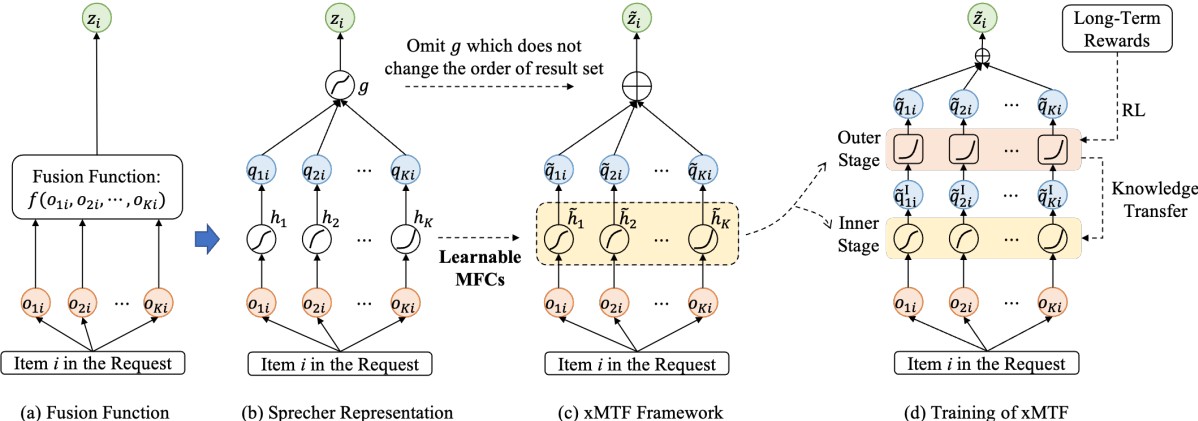

**Figure 2: Overall structure of xMTF.**

**Table 1: Typical fusion formulas in existing research.**

| Index | Formula | Literature |
|---|---|---|
| 1 | $z_i = \sum_{k=1}^{K} a_k o_{ki}$ | [1, 31] |
| 2 | $z_i = \sum_{k=1}^{K} a_k \log(o_{ki} + \beta_k)$ ($\beta_k$ is pre-defined) | [32] |
| 3 | $z_i = \prod_{k=1}^{K} o_{ki}^{a_k}$ | [2] |

to determine the best one, especially since different users might benefit from different formulas. Second, pre-defined formulas restrict the MTF search space, leading to inferior model performance.

To address this issue, this paper provides a *formula-free MTF framework*, called eXtreme MTF (xMTF), as shown in Figure 2. We first show that any suitable fusion function (shown in Figure 2(a)) can be expressed as a combination of single-variable monotonic functions for each prediction $o_{ki}$ (shown in Figure 2(b)). Leveraging this, xMTF introduces a novel monotonic fusion cell (MFC) to model these monotonic transformations of input prediction values, as shown in Figure 2(c). MFCs have the following important features:

- MFCs capture the **monotonicity** between the output and the input variables, which is an important intrinsic structure of MTF, making the MFCs interpretable.
- MFCs are **learnable**, replacing pre-defined fusion formulas in existing methods, thus expanding the MTF search space.
- The learnable MFCs provide **personalized fusion functions** for different users and predictions. In contrast, formula-based approaches can only personalize a few coefficients in pre-defined formulas.

The larger search space complicates xMTF training, and existing RL-based methods are not directly applicable. To overcome this, we provide a two-stage hybrid (TSH) training approach, shown in Figure 2(d), comprising an RL-based outer stage and a supervised-learning-based inner stage. The outer stage contains fewer parameters as RL actions, learning long-term rewards, while the inner stage, with more parameters, learns the knowledge of the outer stage by supervised learning. TSH enables effective xMTF training, showing significant improvement over existing approaches.

In summary, the contribution of this paper is:

- We propose xMTF, a formula-free MTF framework, to expand the search space of MTF. To the best of the authors' knowledge, we are the first to discuss a generalized MTF model rather than formula-based MTF approaches.
- We introduce learnable MFCs to replace pre-defined formulas in MTF. MFCs capture the intrinsic monotonicity of MTF and enable personalized fusion functions, leading to superior performance over formula-based approaches.
- We propose TSH to train xMTF effectively, addressing the challenge of increasing search space.
- Extensive offline and online experiments show the effectiveness of xMTF, and xMTF has been applied to our online system, serving over 100 million users.

## 2 Related Work

### 2.1 Multi-Task Learning

MTL simultaneously predicts multiple user feedback, e.g. click-through rate and like rate. Multi-gate Mixture-of-Experts (MMoE) [17] employs multiple expert networks to capture task-specific patterns while sharing information. Progressive Layered Extraction (PLE) [23] dynamically allocates shared and task-specific layers to enhance accuracy. There is also research addressing negative transfer in embedding learning[21] and gradient conflicts[5, 28, 29].

MTL generates predictions for multiple types of user feedback, but it remains challenging to use these predictions to deliver the final recommendation. To solve this, recommender systems typically employ an MTF module after the MTL module to integrate predictions into a single ranking score.

### 2.2 Multi-Task Fusion

MTF creates a single score for each item to make final recommendations to enhance overall user satisfaction. MTF techniques differ from MTL as users do not give direct feedback on overall satisfaction with each item. Instead, users' overall satisfaction is indicated by long-term rewards. A common approach is to pre-define a fusion formula, e.g. formulas in Table 1, and find optimal coefficients using black-box optimization methods like grid search[12], Bayesian optimization [18], or Cross-Entropy Method (CEM)[20]. These methods yield non-personalized coefficients, limiting their ability to reflect

personal preferences. Recently, RL-based approaches have emerged to provide personalized coefficients, enhancing users' long-term rewards. Han et al. [10] propose a Deep Reinforcement Learning based Ranking Strategy (DRRS) to maximize the platform's cumulative reward by determining personalized coefficients in MTF. Zhang et al. [32] construct the BatchRL-MTF framework for MTF recommendation tasks to address issues like the deadly triad problem and extrapolation error problem of traditional off-policy applied in practical recommender systems. Cai et al. [1] focus on users' retention in RL modeling, while Zhang et al. [31] consider RL-based MTF in multi-stage recommender systems.

However, existing non-RL and RL based MTF approaches rely on pre-defined fusion formulas and search only a limited number of coefficients. The pre-defined formulas restrict the MTF search space, resulting in suboptimal performance. This paper aims to address this limitation by introducing a formula-free MTF framework.

## 3 Problem Formulation

We begin by presenting the RL modeling of MTF, then discuss the monotonicity property and the solutions offered by existing formula-based approaches. Finally, we discuss the challenges to highlight the need for our proposed solution.

### 3.1 RL Modeling of MTF

MTF aims to compute a single merged value for each item based on various predictions to enhance overall user satisfaction. This satisfaction is typically reflected in users' long-term rewards, such as total watch time and retention [1, 31, 32].

Recently, RL-based approaches have been applied to MTF [2, 16, 32]. RL models the interaction between users and recommender systems as a Markov decision process (MDP)[22] with $(\mathcal{S}, \mathcal{A}, \mathcal{P}, R, \rho_0, \gamma)$, where $\mathcal{S}$ is the state space, $\mathcal{A}$ is the action space, $\mathcal{P} : \mathcal{S} \times \mathcal{A} \rightarrow \mathcal{S}$ is the transition function, $R : \mathcal{S} \times \mathcal{A} \rightarrow \mathbb{R}$ is the reward function, $\rho_0$ is the initial state, and $\gamma$ is the discounting factor. As shown in Figure 3, when a user opens the app, a session begins, consisting of multiple requests until the user leaves. At Step $t$, the system obtains a user state $\boldsymbol{s}_t \in \mathcal{S}$ with a candidate set $\Omega_t$ from a candidate retrieval module. The prediction models provide $K$ predictions, i.e. $o_{ki}, 1 \leq k \leq K$, for each item $i \in \Omega_t$. The predictions usually include click-through rate, like rate, expected watch time, etc. Then, the MTF task merges the $K$ predictions into a single score:

$$z_i = f\left(o_{1i}, o_{2i}, \cdots, o_{ki}, \cdots, o_{Ki}; a_t\right), \forall i \in \Omega_t \quad (1)$$

where $a_t \in \mathcal{A}$ is the parameter of $f$, which is the action of RL to be determined. After MTF, the system returns the top items to the user according to the fusion score $z_i$. After watching the recommended items, the user provides feedback $r_t = R\left(\boldsymbol{s}_t, a_t\right)$. Then, the user transfers to the next state $\boldsymbol{s}_{t+1} \sim P\left(\boldsymbol{s}_t, a_t\right)$ and determines whether to send the next request or leave. MTF aims to find the optimal parameter $a_t$ (action of RL) to maximize the long-term reward:

$$\max_{a_t} R_t = \sum_{t'=t}^{T} \gamma^{t'-1} r_{t'} \quad (2)$$

where $T$ is the step that the user leaves the app. The action $a_t$ can be modeled by the policy function $\mu$ from the user state $\boldsymbol{s}_t$:

$$a_t = \mu\left(\boldsymbol{s}_t; \xi\right) \quad (3)$$

where $\xi$ is the parameters. There are many RL models, e.g., DDPG[13], TD3[6], and SAC[9], to get the policy function $\mu$ and the action $a_t$. Therefore, if we provide the parameterized form of the fusion function in Eq. (1) and the parameters $a_t$ to be optimized, typical RL-based approaches can be applied to find the optimal $a_t$. However, the choice of the fusion function $f$ is a nontrivial problem.

### 3.2 Challenges

As discussed, we need a parameterized form of the fusion function $f$ in Eq. (1), and take the parameters $a_t$ as RL actions. In MTF, the fusion function $f$ should have the following monotonicity property.

**Monotonicity Property of the Fusion Function**: the fusion function $f(o_{1i}, o_{2i}, \cdots, o_{Ki})$ is monotonic increasing with respect to each prediction $o_{ki}$. In other words, $\forall k (1 \leq k \leq K)$, if we replace the prediction $o_{ki}$ by $o'_{ki}(o_{ki} < o'_{ki})$, with other input $o_{k'i}, k' \neq k$ unchanged, then we have

$$f(o_{1i}, \cdots, o_{ki}, \cdots, o_{Ki}; a_t) \leq f(o_{1i}, \cdots, o'_{ki}, \cdots, o_{Ki}; a_t)$$

This property is important in MTF because the user feedback modeled in recommender systems, e.g., the click-through rate and the watch time, usually positively relates to the user's satisfaction[26].

To maintain the monotonicity property of the fusion function $f$, existing approaches[1, 31, 32] all consider *formula-based approaches*, i.e., to pre-define a fusion formula with the monotonicity property for $f$, e.g., the formulas in Table 1, and regard the coefficients $a_k$ as RL actions. In such settings, the number of actions equals the number of prediction types, i.e. $K$, which is a very small number, leading to a limited search space and inferior performance. Moreover, the choice of the fusion formulas is also highly uninvestigated.

Can we use a more complicated fusion function $f$ with more parameters to expand the search space? Here, we face two kinds of challenges:

- **Modeling of monotonicity property**. We need to find a suitable model that is sufficiently expressive and capable of capturing the monotonicity property of the fusion function.
- **Training difficulty**. If we use a complicated function $f$, then the parameter $a_t$, which is the RL action, will be very high dimensional, leading to an extremely high training difficulty[34]. Therefore, we need to deal with the trade-off between the search space and the difficulty of training.

To this end, Section 4 provides a *formula-free MTF framework*, called xMTF, to capture the monotonicity property of MTF, while Section 5 discusses the effective training of xMTF.

## 4 The xMTF Framework

We present the xMTF framework, as illustrated in Figure 2. First, we show that the fusion function can be represented as a composition of single-variable monotonic functions for each prediction. Leveraging this, we introduce learnable MFCs to describe these monotonic functions, capturing the fusion function in a very general sense.

### 4.1 Representation of Fusion Function

The key problem is to find a suitable model to capture the monotonicity property of the fusion function. Here we provide a fundamental proposition on the representation of the fusion function by simpler single-variable functions.

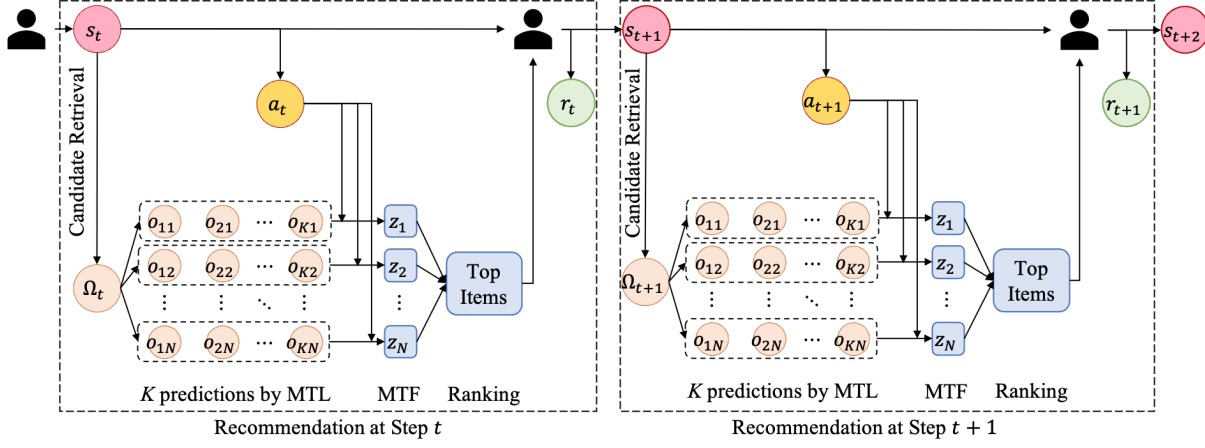

**Figure 3: MDP Modeling of MTF.**

**Table 2: Representation of existing formulas in Table 1.**

| Index in Table 1 | $g$ | $h_k$ |
|---|---|---|
| 1 | $g(x) = x$ | $h_k(o_k) = a_k o_k$ |
| 2 | $g(x) = x$ | $h_k(o_k) = a_k \log(o_k + \beta_k)$ |
| 3 | $g(x) = \exp(x)$ | $h_k(o_k) = a_k \log(o_k)$ |

PROPOSITION 4.1 (REPRESENTATION OF FUSION FUNCTION). *Suppose the fusion function $f(o_{1i}, o_{2i}, \cdots, o_{Ki})$ is monotonic increasing to each $o_{ki}$, then there exists a single-variable monotonic increasing function $g(\cdot)$, and $K$ single-variable monotonic increasing functions $h_k(\cdot), 1 \leq k \leq K$, so that the fusion function $f$ can be represented as*

$$f(o_{1i}, o_{2i}, \cdots, o_{Ki}) = g\left(\sum_{k=1}^{K} q_{ki}\right) \quad (4)$$

*where*

$$q_{ki} = h_k(o_{ki}) \quad (5)$$

PROOF. This proposition is a corollary of the Sprecher Construction [11]. We leave it to Appendix. A. □

This representation is depicted in Figure 2(b). Proposition 4.1 demonstrates that any fusion function can be decomposed into single-variable monotonic functions. This representation unifies and expands existing fusion formulas shown in Table 1. We put the monotonic functions $g$ and $h_k$ of these formulas in Table 2. Different choices of $g$ and $h_k$ result in different fusion formulas.

Proposition 4.1 ensures the existence of monotonic representations for any suitable fusion function. Therefore, by making these monotonic functions learnable, we can significantly expand the search space of MTF, which will be discussed in Section 4.2.

### 4.2 xMTF with Monotonic Fusion Cells

First, note that the increasing function $g$ in Eq. (4) does not need to be modeled, as it does not affect the ordering of the result set (we only need the top items by fusion score!). Therefore, we only need to model the monotonic function $h_k$ for each type of prediction.

Here we define the monotonic fusion cell (MFC) for the $k$-th prediction $o_{ki}$ as a learnable parameterized function with the user state $s_t$ and the prediction $o_{ki}$ as the inputs:

$$\tilde{q}_{ki} = \tilde{h}_k(o_{ki}, s_t; \theta_k) \quad (6)$$

where $\theta_k$ represents the parameters. To ensure the MFC's monotonicity, we use an auxiliary pairwise loss during training:

$$\mathcal{L}_k^{\text{mono}} = \sum_{i,j \in \Omega_t} \mathbf{1}_{o_{ki} < o_{kj}} \max\left\{0, \tilde{h}_k(o_{ki}, s_t; \theta_k) - \tilde{h}_k(o_{kj}, s_t; \theta_k)\right\} \quad (7)$$

where $\mathbf{1}_{o_{ki} < o_{kj}}$ equals 1 when $o_{ki} < o_{kj}$, and equals 0 otherwise.

Given the MFCs $\tilde{q}_{ki}$, we can define a new fusion function $\tilde{f}$ as:

$$\tilde{z}_i = \tilde{f}(o_{1i}, o_{2i}, \cdots, o_{Ki}) = \sum_{k=1}^{K} \tilde{h}_k(o_{ki}, s_t; \theta_k) \quad (8)$$

We have removed the monotonic function $g$ in Eq. (4) as previously discussed. By learning the optimal parameter $\theta_k$, Eq. (8) can generate the final result set without requiring pre-defined fusion formulas used in existing approaches. The model structure of xMTF with MFCs is depicted in Figure 2(c).

Before presenting the training algorithm, we would like to discuss a few points of MFCs.

- MFCs enable **personalized fusion functions**. Compared to the monotonic function in Eq. (5), MFCs in Eq. (6) include the user state $s_t$ as an additional input. If we regard $\tilde{h}_k$ as a monotonic transformation of the prediction variable $o_{ki}$, the input $s_t$ allows for personalized transformation for different requests. Recall that existing approaches focus on personalizing only a few coefficients under pre-defined (non-personalized) fusion formulas. In contrast, MFCs extend the personalized weights to personalized functions.
- As discussed above, MFCs can be interpreted as extensions to the existing fusion formulas in Table 1, which means MFCs capture the **intrinsic monotonicity structure** of MTF. Moreover, this monotonicity of MFCs enables the development of more effective training methods, which will be discussed in Section 5.

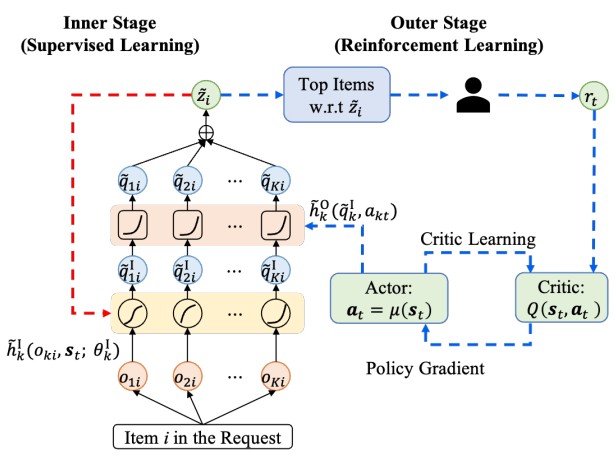

**Figure 4: The TSH training approach.**

## 5 Training of xMTF

This section presents xMTF's training, i.e. to determine $\theta_k, 1 \leq k \leq K$ in Eq. (6). We propose a novel two-stage hybrid (TSH) training approach, illustrated in Figure 4, to tackle the training difficulty, which contains an RL stage with a few actions and a supervised knowledge transfer stage with more learnable parameters.

### 5.1 Overall Framework

Recall that our objective is to maximize the long-term user experience $R_t$ in Eq. (2). Therefore, we use the RL-based approach, as illustrated in Section 3. To reduce the difficulty of RL training, we must decrease the number of actions in RL. Thus, we cannot treat all $\theta_k, 1 \leq k \leq K$ as actions, since each $\theta_k$ includes many parameters.

Here, we develop a novel two-stage hybrid (TSH) training approach, as shown in Figure 4. The key idea is to divide the MFCs into two cascade monotonic parts. Specifically, we decompose the parameter $\theta_k$ into two parts by defining $\theta_k = (\theta_k^{\mathrm{I}}, a_k)$, and separate the MFCs in Eq. (6) into two monotonic stages:

$$\tilde{q}_{ki}^{\mathrm{I}} = \tilde{h}_k^{\mathrm{I}}\left(o_{ki}, \boldsymbol{s}_t; \theta_k^{\mathrm{I}}\right) \tag{9}$$

$$\tilde{q}_{ki} = \tilde{h}_k^{\mathrm{O}}\left(\tilde{q}_{ki}^{\mathrm{I}}; a_k\right) \tag{10}$$

We denote $\tilde{h}_k^{\mathrm{I}}$ as the inner stage and $\tilde{h}_k^{\mathrm{O}}$ as the outer stage. The primary difference between the two stages is the number of parameters. The inner stage $\tilde{h}_k^{\mathrm{I}}$ includes a large number of parameters in $\theta_k^{\mathrm{I}}$ to ensure sufficient expressiveness, whereas the outer stage $\tilde{h}_k^{\mathrm{O}}$ only includes a few parameters in $a_k$. Note that choosing a simple function for $\tilde{h}_k^{\mathrm{O}}$ does not reduce the expressiveness of MFCs. This is because, given any original function $\tilde{h}_k$, we can find a corresponding $\tilde{h}_k^{\mathrm{I}}$ to satisfy Eq. (9)(10). Actually, we just need to set $\tilde{h}_k^{\mathrm{I}} = \alpha\left(\tilde{h}_k\right)$, where $\alpha(\cdot)$ is the inverse function of $\tilde{h}_k^{\mathrm{O}}$ w.r.t. $\tilde{q}_{ki}^{\mathrm{I}}$:

We train the two stages by different methods. For the outer stage with a few parameters $(a_k)$, we use RL-based approaches to optimize the long-term user experience. In contrast, for the inner stage with many parameters $(\theta_k^{\mathrm{I}})$, we apply supervised learning to ensure good convergence. However, the supervised learning of the inner stage is still challenging since the labels needed for learning are not directly

provided by users (note that the users do not provide integrated feedback for each item to guide the learning of $\tilde{h}_k^{\mathrm{I}}$!)

We adopt knowledge transfer to overcome this. We use the output of the outer stage as the label for the inner stage, as shown in Figure 4. In this setup, **the outer stage continuously adjusts the inner stage's output based on long-term user experience, while the inner stage absorbs new knowledge from the outer stage**. This method allows the inner stage to learn about long-term user satisfaction without directly using RL-based techniques.

Now we are ready to provide the final training algorithm of xMTF, as shown in Algorithm 1, of which the details are provided in the next two subsections.

### 5.2 Outer Stage

The outer stage is a simple parameterized function $\tilde{h}_k^{\mathrm{O}}$ with a few parameters $a_k$. In practice, we just use a second-order function:

$$\tilde{h}_k^{\mathrm{O}}\left(\tilde{q}_{ki}^{\mathrm{I}}; a_k\right) = \tilde{q}_{ki}^{\mathrm{I}}\left(1 + a_k \tilde{q}_{ki}^{\mathrm{I}}\right) \tag{11}$$

In this case, $a_k$ is a scalar for each $k$.

We regard $a_k$ as the RL action to be optimized. The choice of the specific RL algorithm is orthogonal to the contribution of this paper, and we adopt a typical actor-critic structure. Specifically, we define the action vector $\boldsymbol{a} = \{a_1, \cdots, a_K\}$, and the actor network generates the action $\boldsymbol{a}$ by a parameterized function:

$$\boldsymbol{a}_t = \mu\left(\boldsymbol{s}_t; \xi\right) \tag{12}$$

where the subscript $t$ in $\boldsymbol{a}_t$ means the action depends on the time step $t$, and $\xi$ is the parameter.

Then we define the critic function $Q\left(\boldsymbol{s}_t, \boldsymbol{a}_t; \phi\right)$ to estimate the long-term reward $R_t$. The loss of critic learning is

$$\mathcal{L}^{\mathrm{critic}}(\phi) = \left[Q\left(\boldsymbol{s}_t, \boldsymbol{a}_t; \phi\right) - \left(r_t + \gamma Q\left(\boldsymbol{s}_{t+1}, \mu\left(\boldsymbol{s}_{t+1}; \xi^-\right); \phi^-\right)\right)\right]^2 \tag{13}$$

where $\xi^-$ is the parameter of the target actor, and $\phi^-$ is the parameter of the target critic. Furthermore, the policy gradient of parameter $\xi$ is calculated by

$$\nabla_\xi J = \nabla_\xi \mu\left(\boldsymbol{s}_t; \xi\right) \nabla_\mu Q\left(\boldsymbol{s}_t, \mu\left(\boldsymbol{s}_t; \xi\right)\right) \tag{14}$$

### 5.3 Inner Stage

The inner stage aims to learn from the outer stage to capture long-term user experience. We define the sum of the inner stage as:

$$\tilde{z}_i^{\mathrm{I}} = \sum_{k=1}^K \tilde{q}_{ki}^{\mathrm{I}} \tag{15}$$

Compared to the actual output $\tilde{z}_i$ defined in Eq. (8), the sum $\tilde{z}_i^{\mathrm{I}}$ excludes the outer stage $\tilde{h}_k^{\mathrm{O}}$. Then, we use $\tilde{z}_i$ to guide the learning of $\tilde{z}_i^{\mathrm{I}}$, to transfer the knowledge from the outer stage to the inner stage. Specifically, we define a pairwise loss, i.e., the BPR loss[19], to ensure the consistency of the ranking order between $\tilde{z}_i^{\mathrm{I}}$ and $\tilde{z}_i$:

$$\mathcal{L}^{\mathrm{transfer}} = -\sum_{i,j \in \Omega_t} \mathbf{1}_{\tilde{z}_i < \tilde{z}_j} \log \sigma(\tilde{z}_j^{\mathrm{I}} - \tilde{z}_i^{\mathrm{I}}) \tag{16}$$

where the $\tilde{z}_i$ is regarded as the label, and $\sigma(\cdot)$ is the sigmoid function. Eq. (16) means that the inner stage aims to absorb the knowledge from the outer stage, which is achievable because the inner stage

---

**Algorithm 1** The training process of xMTF.

1: Input: training data (replay buffer) $\{s_{1:T}, a_{1:T}, r_{1:T}, o_{1:T}\}$ for each user.
2: Output: The inner stage $\tilde{h}_k^{\mathrm{I}}\left(o_{ki}, s_t; \theta_k^{\mathrm{I}}\right)$ of MFCs, parameterized by $\theta_k^{\mathrm{I}}$; a policy $\mu\left(s_t; \xi\right)$ used in the outer stage of MFCs, parameterized by $\xi$; a critic function $Q\left(s_t, a_t; \phi\right)$ parameterized by $\phi$.
3: **for** each user session with $T$ requests from the replay buffer **do**
4:     **for** $t = 1, \cdots, T$ **do**
5:         **Data preparation**: Collect the reward $r_t$, the predictions $o_t$ and the action $a_t$ from the replay buffer.
6:         **Critic learning of the outer stage**: $\phi \leftarrow \phi - \alpha \nabla_\phi \mathcal{L}^{\mathrm{critic}}(\phi)$, where $\mathcal{L}^{\mathrm{critic}}(\phi)$ is defined in Eq. (13), and $\alpha$ is the learning rate.
7:         **Actor learning of the outer stage**: $\xi \leftarrow \xi - \beta \nabla_\xi J$, where $\nabla_\xi J$ is defined in Eq. (14), and $\beta$ is the learning rate.
8:         **Learning of the inner stage**: $\theta_k^{\mathrm{I}} \leftarrow \theta_k^{\mathrm{I}} - \eta \nabla_{\theta_k^{\mathrm{I}}} \mathcal{L}^{\mathrm{I}}(\theta_k^{\mathrm{I}})$, where $\mathcal{L}^{\mathrm{I}}(\theta_k^{\mathrm{I}})$ is defined in Eq. (18), and $\eta$ is the learning rate.
9:     **end for**
10: **end for**

---

has significantly greater expressiveness than the outer stage. Consequently, in TSH, the outer and inner stages continuously interact: the outer stage adjusts the fusion results to align with long-term user experience, while the inner stage continuously learns from the outer stage.

Moreover, we re-define the monotonicity loss in Eq. (7) for the inner stage:

$$\mathcal{L}_k^{\mathrm{mono,I}} = \sum_{i,j \in \Omega_t} \mathbf{1}_{o_{ki} < o_{kj}} \max\left\{0, \tilde{h}_k^{\mathrm{I}}\left(o_{ki}, s_t; \theta_k^{\mathrm{I}}\right) - \tilde{h}_k^{\mathrm{I}}\left(o_{kj}, s_t; \theta_k^{\mathrm{I}}\right)\right\}$$
(17)

The final loss of the inner stage is

$$\mathcal{L}^{\mathrm{I}} = \lambda \sum_{k=1}^{K} \mathcal{L}_k^{\mathrm{mono,I}} + (1 - \lambda)\mathcal{L}^{\mathrm{transfer}}$$
(18)

where $\lambda$ is the hyper-parameter to balance the final loss.

## 6 Experiments

We consider the following research questions (RQs):

- **RQ1**: How does xMTF perform compared to other state-of-the-art MTF methods?
- **RQ2**: Does MFCs improve the performance of xMTF? Can we interpret the outputs of MFCs?
- **RQ3**: How does each part of the TSH training method affect the performance of xMTF?
- **RQ4**: Can xMTF improve the performance of MTF tasks in real-world online recommender systems?

### 6.1 Offline Experiment Settings

6.1.1 *Dataset and Metrics.* We choose KuaiRand [7] as the offline experimental dataset. This public dataset from Kuaishou contains 27,285 users and 32,038,725 items, providing contextual features of users and items, along with various user feedback signals. We consider six types of user feedback: click, long view, like, follow, comment, and share, and the statistics of the feedback are shown in Table 6 in Appendix. B. The performance of the MTL model is not the main focus of this paper, so we use MMoE[17], a widely-used MTL model, to generate the predictions for user feedback on each item. Once predictions are obtained, MTF merges them into a single

final score and then returns the top items to the user. This section investigates the performance of different MTF methods[1].

To emulate user behavior upon receiving the recommended items, we construct an offline simulator to act as the environment, simulating user interaction with the recommender system. When the simulator receives recommended items for the current request, it generates user feedback and decides whether to send the next request. We define the following exit rules [27]: if users have exhausted their satisfaction, they will leave the session. An early exit may lead to worse recommendation performance.

In the abovementioned experimental settings, we choose a long-term reward, i.e., the **Total Watch Time** of all the items in a complete session, as the evaluation metric.

6.1.2 *Details.* The user state $s_t$ includes the user profile, the behavior history, and the request context. In xMTF framework, for the inner stage, we use a multi-layer perception (MLP) to model the function $\tilde{h}_k^{\mathrm{I}}\left(o_{ki}, s_t; \theta_k^{\mathrm{I}}\right)$ defined in Eq. (9), where $\theta_k^{\mathrm{I}}$ are the parameters in MLP; for the outer stage, we use RL to model the function $\tilde{h}_k^{\mathrm{O}}\left(\tilde{q}_{ki}^{\mathrm{I}}; a_k\right)$ defined in Eq. (10), where $a_k$ are the actions of RL. To ensure fairness, we keep the same network architecture for the actors and critics across all compared methods, which consists of a five-layer MLP. The detailed hyper-parameters are shown in Table 7 in Appendix. B. For each experiment, we conduct 20 trials to calculate the mean performance and standard deviations.

6.1.3 *Baselines.*

- **Cross Entropy Method (CEM)** [20]: a black-box optimization method commonly used for hyper-parameter optimization. CEM searches the (non-personalized) optimal parameters in pre-defined fusion formulas. Two kinds of formulas are tested with CEM.
  - **CEM-1**: Adopt the first formula in Table 1, i.e., $z_i = \sum_{k=1}^{K} a_k o_{ki}$, where $a_k$ is the parameter to be optimized.
  - **CEM-2**: Adopt the second formula in Table 1, i.e., $z_i = \sum_{k=1}^{K} a_k \log\left(o_{ki} + \beta_k\right)$, where $a_k$ is the parameter to be optimized.
- **TD3** [6]: TD3 searches the personalized optimal parameters in pre-defined fusion formulas. Like CEM, we apply two kinds of formulas to TD3, called **TD3-1** and **TD3-2**.

---

[1]The code can be referred to at https://anonymous.4open.science/r/xMTFwww2025-BC2E/

**Table 3: The offline performance of different methods.**

| Methods | Total Watch Time (s) |
|---|---|
| CEM-1 | 897.5(±12.3) |
| CEM-2 | 931.2(±13.5) |
| TD3-1 | 1088.7(±13.7) |
| TD3-2 | 1129.1(±14.8) |
| BatchRL-MTF-1 | 1137.3(±13.1) |
| BatchRL-MTF-2 | 1185.4(±12.6) |
| **xMTF** | **1279.7(±12.9)** |
| xMTF w/o outer stage | 1092.8(±9.1) |
| xMTF w/o inner stage | 1106.3(±11.2) |

- **BatchRL-MTF** [32]: A recent RL method for formula-based MTF in recommender systems. Like CEM, we apply two kinds of formulas to BatchRL-MTF, called **BatchRL-MTF-1** and **BatchRL-MTF-2**.
- **xMTF**: our proposed formula-free MTF framework which does not rely on a pre-defined formula.
- **xMTF w/o outer (inner) stage**: xMTF without the outer (inner) stage in TSH, which will be discussed in Section 6.4.

## 6.2 Performance Comparison (RQ1)

Table 3 shows the results of different methods. CEM searches global coefficients for all users, performing significantly worse than RL-based methods. This highlights the importance of providing personalized coefficients for different users and considering long-term user satisfaction to enhance MTF performance. For RL-based methods like TD3, the performance of TD3-2 is better than that of TD3-1, showing that the fusion formulas do affect MTF performance. The proposed xMTF achieves the best performance among all methods because it does not rely on a pre-defined formula and provides a larger search space for better performance.

## 6.3 Impacts of MFCs (RQ2)

*6.3.1 Monotonicity of MFCs.* We visualize the MFCs in Figures 5 and 6. Figure 5 plots the outputs of the MFC ($\tilde{q}_{ki}$), in relation to the inputs of MFC ($o_{ki}$) for different users, where $k$ is the subscript corresponding to the prediction of the "long view" behavior. Figure 6 plots the outputs and inputs of the MFC of a certain user across different predictions. Evidently, all functions exhibit monotonic properties, consistent with the fact that the fusion function should be monotonic with respect to the input variables. This monotonicity is, of course, a result of the monotonicity loss in Eq. (17).

*6.3.2 Personalized Fusion Functions Provided by MFCs.* MFCs capture different monotonic properties for different users and predictions. Specifically, Figure 5 shows the MFCs learned for different users are different, while Figure 6 shows the MFCs learned for different predictions are also different. These personalized fusion functions are a direct result of the larger search space introduced by the learnable MFCs, which is a key reason why xMTF outperforms existing methods that rely on predefined formulas.

*6.3.3 Impacts of the Monotonicity Loss on the Performance.* We have shown that MFCs capture monotonicity, but does the monotonicity have a real impact on performance? To illustrate this, we conduct experiments under different hyper-parameter $\lambda$ in Eq. (18),

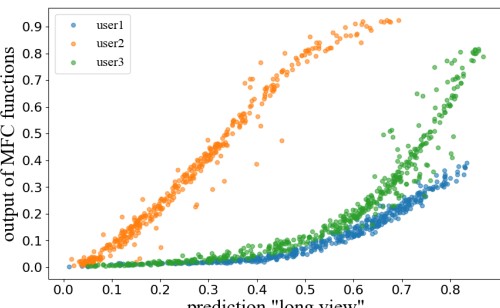

**Figure 5: The MFC functions with respect to different users of the same prediction "long view".**

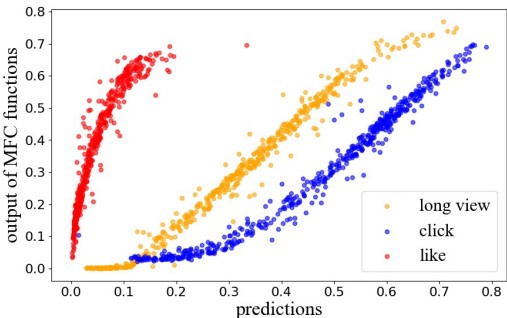

**Figure 6: The MFC funtions with respect to different predictions of the same user.**

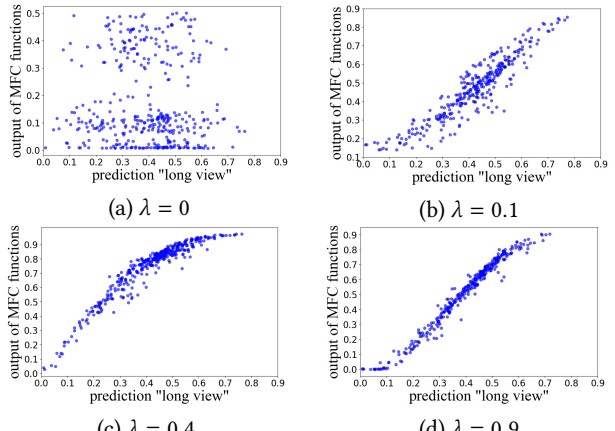

(a) $\lambda = 0$          (b) $\lambda = 0.1$

(c) $\lambda = 0.4$          (d) $\lambda = 0.9$

**Figure 7: The MFC functions with respect to prediction "long view" and different hyper-parameters $\lambda$.**

**Table 4: Performance under different hyper-parameter $\lambda$.**

| Hyper-Parameter $\lambda$ | Total Watch Time (s) |
|---|---|
| $\lambda = 0$ | 732.8(±18.2) |
| $\lambda = 0.1$ | 1073.2(±14.6) |
| $\lambda = 0.4$ | 1279.7(±12.9) |
| $\lambda = 0.9$ | 1254.4(±12.5) |
| $\lambda = 1$ | 1103.1(±16.3) |

which describes the importance of monotonicity loss $\mathcal{L}^{\text{mono,I}}$ defined in Eq. (17). $\lambda = 0$ means that we do not apply any monotonicity loss for xMTF, while $\lambda = 1$ indicates that we focus solely on the

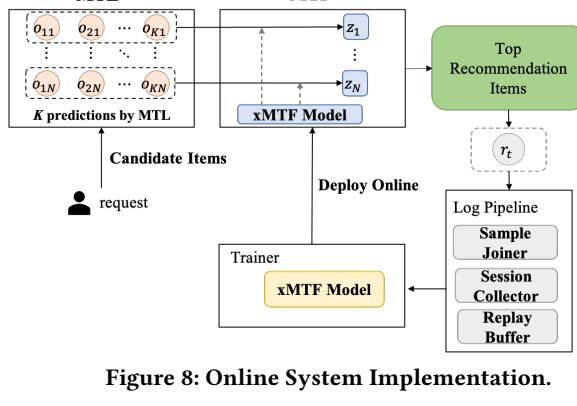

**Figure 8: Online System Implementation.**

monotonicity loss. Figure 7 shows the outputs of MFCs w.r.t. the predictions of "long view" under different values of $\lambda$. Figure 7(a) shows no monotonicity between the inputs and the outputs of the MFCs under $\lambda = 0$. Table 4 shows that the performance when $\lambda = 0$ is much worse than those in other experiments, highlighting the necessity of the monotonicity captured by MFCs. $\lambda = 0.4$ achieves the best performance in Table 4, and it is the hyper-parameter we ultimately selected. Compared with $\lambda = 0.1$, $\lambda = 0.4$ exhibits better monotonicity, as shown in Figures 7(b)(c), thus achieving a better result. $\lambda = 0.9$ shows a little better monotonicity than $\lambda = 0.4$, as shown in Figures 7(c)(d), but it weakens the effect of supervised loss $\mathcal{L}^{\text{transfer}}$ which reflects the user satisfaction. Therefore, $\lambda = 0.9$ also leads to a worse result than $\lambda = 0.4$. In addition, $\lambda = 1$ achieves a worse result due to the absence of supervised loss $\mathcal{L}^{\text{transfer}}$.

### 6.4 Impacts of the Training Method (RQ3)

This subsection provides the ablation studies to show the effectiveness of the inner and outer stages in the proposed TSH method.

Firstly, we remove the outer stage while retaining the inner stage (denoted by **xMTF w/o outer stage**), meaning we do not use RL to model long-term user satisfaction. The total watch time, shown in Table 3 is 1092.8 s, much worse than the result of xMTF, demonstrating the necessity of considering long-term user satisfaction.

Moreover, we remove the inner stage while retaining the outer stage (denoted by **xMTF w/o inner stage**), meaning we remove the most expressive component of xMTF. In this scenario, xMTF actually degenerates into a formula-based MTF model with the formula $z_i = \sum_{k=1}^{K} o_{ki}(1 + a_k o_{ki})$. As shown in Table 3, the total watch time is 1106.3 s, worse than xMTF, highlighting the advantages of our proposed xMTF model over a formula-based MTF model.

### 6.5 Online Experiments (RQ4)

We assess the performance of xMTF on a popular short video platform with over 100 million users. This platform considers user feedback types such as effective view, long view, complete playback, expected watch time, like, follow, comment, share, and profile visit for each item. The MTF on this platform combines predictions of these feedback types into a final score to select videos with the highest user satisfaction. We adopt users' **Daily Watch Time** as the evaluation metric because it serves as a long-term reward, reflecting overall user satisfaction with all recommended videos. This

**Table 5: The online performance of xMTF.**

|  | Performance Gain of **xMTF** |
|---|---|
| Daily Watch Time | **+0.833%** [-0.11%, 0.11%] |
| Play Counts | **+0.583%** [-0.14%, 0.14%] |
| Comment | **+2.391%** [-1.26%, 1.26%] |
| Share | **+2.205%** [-0.81%, 0.81%] |

metric is also widely used in existing research [3, 30, 31]. Besides daily watch time, we also examine play counts and interactions.

The xMTF structure in the online system is depicted in Figure 8. The xMTF model is continuously trained in a streaming manner. Specifically, when a user exits a session, the session's data is immediately sent to the xMTF model for training, and the updated model is deployed online for the MTF task in recommender systems. In practice, the xMTF model converges within two days when trained from scratch, and it is then continuously trained and updated online, serving users on the short video platform.

In the online experiments, we randomly divide users into two equal-sized groups: the baseline group and the experimental group. If the user experience of the experimental group, as indicated by evaluation metrics, surpasses that of the baseline group, we replace the baseline group's models with those from the experimental group, establishing the latter as the new baseline. Using this experimental setup, we sequentially tested CEM[20], TD3[6], UNEX-RL[31], and xMTF in our scenario. The most recent baseline before deploying xMTF was UNEX-RL, a state-of-the-art formula-based MTF model considering multiple stages in recommender systems. The performance of UNEX-RL over the previous baselines (CEM, TD3) is shown in Appendix C. Here, we focus on discussing the performance gain of xMTF over the most recent baseline, UNEX-RL.

We evaluated the performance gain of xMTF over the baseline model for 7 consecutive days. Table 5 shows the performance gains of xMTF over the baseline, along with confidence intervals. xMTF achieves significantly better performance, with a 0.833% increase in daily watch time and notable improvements in other evaluation metrics. The changes in some interaction metrics, e.g., like, follow, do not exceed the confidence intervals, which are not listed in Table 5. It is important to note that a 0.1% improvement in daily watch time is statistically significant on our platform. A 0.833% improvement has been one of the largest improvements this year[2]. These online experiments demonstrate the effectiveness of xMTF.

### 7 Conclusion

This paper proposes a formula-free multi-task fusion (MTF) framework, called eXtreme MTF (xMTF), for maximizing long-term user satisfaction in recommender systems. The xMTF framework utilizes a monotonic fusion cell (MFC) to capture the monotonicity property of MTF, eliminating the need for pre-defined formulas. To address the training challenges posed by the larger search space in the xMTF framework, a two-stage hybrid (TSH) learning method is developed to train xMTF effectively. Extensive offline and online experiments demonstrate the effectiveness and advantages of the xMTF framework compared to existing formula-based MTF methods. The xMTF model has been fully deployed in our online system, serving over 100 million users.

---

[2]Please refer to Appendix C for performance gains from previous MTF experiments.

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

# A    Proof of Proposition 4.1

We first quota the Sprecher Representation Theorem[11]:

THEOREM A.1 (SPRECHER REPRESENTATION THEOREM). *For any integer $K \geq 2$ and a domain $I$, there exists real-valued, strictly monotone functions $h_k(x)$, $1 \leq k \leq K$, so that*

(1) *The function $y = h(x_1, x_2, \cdots, x_K) = \sum_{k=1}^{K} h_k(x_k)$ is a bijection between $y \in I$ and $(x_1, x_2, \cdots, x_K) \in I^K$, and when $y$ increases, each $x_k$ will not decrease. We call the $h_k$ the Sprecher construction.*

(2) *Any continuous function of n variables $f(x_1, x_2, \cdots, x_n)$ with domain $[0, 1]^K$ can be represented in the form*

$$f(x_1, x_2, \cdots, x_n) = g(y) = g\left[ \sum_{k=1}^{K} h_k(x_k) \right] \tag{19}$$

*with a (usually non-continuous) function $g$.*

PROOF OF PROPOSITION 4.1.    We only need to prove the monotonicity of the function $g$, and the only extra feature to be considered is the monotonicity property of the function $f$. Specifically, as $y$ increases, each $x_k$ will not decrease. According to the monotonicity property of $f$, this ensures that $g(y) = f(x_1, x_2, \cdots, x_n)$ is non-decreasing. □

# B    Data and Hyper-Parameters

**Table 6: The statistics of user feedback in KuaiRand.**

| Feedback | Sparse Ratio |
|---|---|
| click | 37.93% |
| long view | 26.35% |
| like | 1.51% |
| follow | 0.12% |
| comment | 0.25% |
| share | 0.09% |

**Table 7: The hyper-parameters of xMTF.**

| Hyper-parameter | Value |
|---|---|
| Optimizer | Adam |
| Actor Learning Rate | 0.0001 |
| Critic Learning Rate | 0.0002 |
| Supervised Learning Rate | 0.0002 |
| Action Dimensions of Actor | 6 |
| Discount Factor | 0.9 |
| Replay Buffer Size | $1 * 10^6$ |
| Train Batch Size | 1024 |
| Fine-Tuning | True |
| Normalized Observations | True |
| Training Platform | Tensorflow |

# C    Previous Baselines

CEM was our earliest baseline model, and we sequentially deployed TD3, and UNEX-RL in our online system. The online performance gain of TD3 over CEM is shown in Table 8, and the online performance gain of UNEX-RL over TD3 is shown in Table 9.

Received 14 October 2024

**Table 8: The online performance gain of TD3 over CEM.**

|  | Performance Gain of **TD3** |
|---|---|
| Daily Watch Time | **+0.414%** [-0.12%, 0.12%] |

**Table 9: The online performance gain of UNEX-RL over TD3.**

|  | Performance Gain of **UNEX-RL** |
|---|---|
| Daily Watch Time | **+0.556%** [-0.11%, 0.11%] |

