# OpenReview forum: "xMTF: A Formula-Free Model for Reinforcement-Learning-Based Multi-Task Fusion in Recommender Systems"
_ACM.org/TheWebConf/2025/Conference — WWW 2025 Poster_

### Official Review · Reviewer_g2tD · 2024-11-19

**Novelty:** 5
**Technical Quality:** 5

**Review:**

In this paper, the authors study an important problem, i.e., modeling multiple types of user feedback in a recommender system. In particular, the authors focus on the multi-task fusion part (shown in Figure 1), and design a learnable (rather than pre-defined) monotonic fusion cell (MFC), which is illustrated in Figure 2(c). Moreover, the authors propose a two-stage training strategy to learn the model parameters effectively shown in Figure 2(d). Empirical studies on both offline and online experiments show the effectiveness of the proposed method (xMTF) over the adopted baseline methods. Overall, the paper is well presented.

**Questions:**

1 The authors shall provide time complexity analysis and time cost results for the MFC part and the TSH training strategy (in the context of the compared baseline methods).

2 The baselines are too few (only 3), and the most recent one is published in 2022. There are lots of existing works on modeling multiple types of user feedback in a recommender system. The authors shall include more recent baselines in the experiments.

**Reviewer Confidence:**

3: The reviewer is confident but not certain that the evaluation is correct

**Scope:**

4: The work is relevant to the Web and to the track, and is of broad interest to the community

---

### Official Review · Reviewer_ZvK5 · 2024-11-28

**Novelty:** 5
**Technical Quality:** 5

**Review:**

**Strengths**:
- This paper solves an important problem in the recommendation scenario, how to merge the results of multiple predictions to output the final recommended items.
- The writing and structure of this paper are very clear, including motivation, methods and verification.
- This paper conducts a lot of experiments in offline and online scenarios to verify the effectiveness of the method.

**Weaknesses**:
- Since this paper adopts a two-stage approach, it is not clear what the time complexity of the proposed method is. First, how long is the offline training phase compared to the baseline? Second, after xMTF is deployed online, what is the inference latency compared to not using the xMTF strategy?
- In the visualization of Section 6.3.2, what are the selection criteria for different users and different predictions? Especially in the former, are there three users randomly selected? More diverse user selection criteria can enhance persuasiveness.
- In Table 4, we noticed that the hyperparameter $\lambda$ has a great impact on the results. How to quickly determine the appropriate hyperparameter for new data sets/application scenarios? Does the author have any suggestions for this?
- In Table 5, the proposed xMTF has obvious differences in the improvement of different targets. What is the difference in the impact of xMTF on different targets? Does the author have any deeper explanation?

**Questions:**

Refer to the Weaknesses section

**Reviewer Confidence:**

2: The reviewer is willing to defend the evaluation, but it is likely that the reviewer did not understand parts of the paper

**Scope:**

4: The work is relevant to the Web and to the track, and is of broad interest to the community

---

### Official Review · Reviewer_9VmM · 2024-12-02

**Novelty:** 5
**Technical Quality:** 5

**Review:**

The paper introduces a new model called xMTF, which is designed to address the pre-defined formula restriction of MTF.  xMTF doesn't rely on fixed formulas and introduces a novel learnable monotonic fusion cell (MFC) and corresponding learning strategy to replace pre-defined formulas. Extensive offline and online experiments show the effectiveness of xMTF.

**Questions:**

1. The baselines are not the latest, the most recent one was published in 2022. Besides, the total citations seem not sufficient?
2. There could be more background introduction for non-experts in recommender systems, as this paper discussed a detailed technique in this area.
3. Will the performance of xMTF change as the types of user feedback change or the user group change? There should be more related experiments or discussions.
4. I am not sure if this paper is better for the industry track?

**Reviewer Confidence:**

2: The reviewer is willing to defend the evaluation, but it is likely that the reviewer did not understand parts of the paper

**Scope:**

3: The work is somewhat relevant to the Web and to the track, and is of narrow interest to a sub-community

---

### Official Review · Reviewer_EBYW · 2024-12-02

**Novelty:** 6
**Technical Quality:** 6

**Review:**

### summary：

This paper focuses on studying recommender systems handling multiple types of feedback. Specifically, it introduces a novel approach to multi-task fusion xMTF in recommender systems, addressing the limitations of existing RL-based methods, which rely on pre-defined formulas. These formulas restrict the RL search space and hinder MTF performance. Leveraging the Sprecher Representation Theorem, this paper demonstrate that any suitable fusion function can be expressed as a composition of single-variable monotonic functions and it further introduces a learnable monotonic fusion cell, replacing the need for pre-defined formulas. Additionally, a two-stage hybrid learning strategy is employed to train xMTF effectively. Experimental results show that xMTF outperforms existing methods in offline settings, and achieves real-world improvements in an online setting on a short video platform.

### strengths:

* **S1**：This paper is well-written, with clear and concise figures and tables, and the methods are simple yet effective. Even readers unfamiliar with reinforcement learning can easily grasp the motivation and contributions of this paper.
* **S2**：The formula-free MTF module proposed in this paper is novel and promising. By analyzing the monotonicity requirements at the variable level of the multi-task fusion function, the paper uses Sprecher construction to decompose it into the sum of monotonic fusion functions for each task, which is then input into an overall monotonic fusion function (preserving rankings, thus can be omitted).
* **S3**：The offline dataset and online platform experiments in this paper effectively validate the performance of xMTF. Subsequent ablation studies demonstrate the necessity of the monotonicity constraint. The visualization of the monotonic fusion cell also confirms its adherence to monotonicity.

### weaknesses：

* I personally believe there are no significant shortcomings in this paper. However, I do lack a background in reinforcement learning.

**Questions:**

### Question：

* **Q1:**  Due to the learnable MFC replacing the heuristic formula, the network parameters have increased. How does the training and inference efficiency of xMTF compare to the baselines?
* **Q2:** The appendix shows the technical iteration path in practical applications (CEM → TD3 → UNEX-RL → xMTF), with gradual improvements in the Daily Watch Time metric. How does the time consumption of the methods along this path compare?

**Reviewer Confidence:**

3: The reviewer is confident but not certain that the evaluation is correct

**Scope:**

4: The work is relevant to the Web and to the track, and is of broad interest to the community

---

### Official Review · Reviewer_DDQq · 2024-12-04

**Novelty:** 4
**Technical Quality:** 4

**Review:**

The submission proposes a new reinforcement learning methods which proposes to replace the predefined formulas which is claimed to restrict the search space of the system.

The submission is well written, and I believe its clear what they are doing. However I struggle to see whats the difference between this solution and a contextual bandit. The authors claim to optimize for long term metrics, but only looks at a session, while I believe long term metrics have a bit longer perspective such as days or weeks even months.

The authors also claim that they enable the system to output more diverse recommendations without showing any evidence of this.

**Questions:**

Thank you for your submission it was interesting to read. However, I have a few questions I think are unclear.

Using functions or models to decide on the weights in the ranking layer has been done in many other scenarios under the name of contextual bandits. Could you explain the difference between those and your solution?

The abstract states that other solutions use "pre-defined formulas that restrict the RL search and become a bottleneck for MTF." could you explain how this solution provides more diverse recommendations and how you have measured it?

Long-term metrics are measured over days or even months, not just how long the session is. Does this system support this, and how has it been shown that this solution improves it? The evaluation seems to optimize for short-term metrics.

**Reviewer Confidence:**

3: The reviewer is confident but not certain that the evaluation is correct

**Scope:**

3: The work is somewhat relevant to the Web and to the track, and is of narrow interest to a sub-community